# Optogenetic control of excitatory post-synaptic differentiation through neuroligin-1 tyrosine phosphorylation

Mathieu Letellier[1,2], Matthieu Lagardère[1,2], Béatrice Tessier[1,2], Harald Janovjak[3,4], Olivier Thoumine[1,2]*

[1]University of Bordeaux, Interdisciplinary Institute for Neuroscience, UMR 5297, Bordeaux, France; [2]CNRS, Interdisciplinary Institute for Neuroscience, UMR 5297, Bordeaux, France; [3]Australian Regenerative Medicine Institute (ARMI), Faculty of Medicine, Nursing and Health Sciences, Monash University, Clayton, Australia; [4]European Molecular Biology Laboratory Australia (EMBL Australia), Monash University, Clayton, Australia

**Abstract** Neuroligins (Nlgns) are adhesion proteins mediating trans-synaptic contacts in neurons. However, conflicting results around their role in synaptic differentiation arise from the various techniques used to manipulate Nlgn expression level. Orthogonally to these approaches, we triggered here the phosphorylation of endogenous Nlgn1 in CA1 mouse hippocampal neurons using a photoactivatable tyrosine kinase receptor (optoFGFR1). Light stimulation for 24 hr selectively increased dendritic spine density and AMPA-receptor-mediated EPSCs in wild-type neurons, but not in Nlgn1 knock-out neurons or when endogenous Nlgn1 was replaced by a non-phosphorylatable mutant (Y782F). Moreover, light stimulation of optoFGFR1 partially occluded LTP in a Nlgn1-dependent manner. Combined with computer simulations, our data support a model by which Nlgn1 tyrosine phosphorylation promotes the assembly of an excitatory post-synaptic scaffold that captures surface AMPA receptors. This optogenetic strategy highlights the impact of Nlgn1 intracellular signaling in synaptic differentiation and potentiation, while enabling an acute control of these mechanisms.

*For correspondence:
olivier.thoumine@u-bordeaux.fr

**Competing interests:** The authors declare that no competing interests exist.

## Introduction

How early neuronal connections mature into functional synapses is a key question in neurobiology, and adhesion molecules such as neuroligins (Nlgns) are thought to play important roles in this process (*Bemben et al., 2015b*; *Craig and Kang, 2007*; *Südhof, 2008*). However, there is an ongoing controversy about the function of Nlgns in synaptic differentiation, arising from divergent results obtained using knock-out (KO), knockdown (KD), and overexpression (OE) approaches. Specifically, whereas Nlgn OE or KD bi-directionally affect synapse number, full or conditional Nlgn1/2/3 KO does not alter synapse density (*Chanda et al., 2017*; *Chih et al., 2005*; *Levinson et al., 2005*; *Prange et al., 2004*; *Varoqueaux et al., 2006*), suggesting that Nlgns are not generally required for synaptogenesis. To address this apparent conflict, experiments that mixed wild type and Nlgn1 KO neurons suggested the interesting model that neurons might compete with one another for synapse formation, depending on their intrinsic Nlgn1 level (*Kwon et al., 2012*).

Besides the role of Nlgns in controlling synapse number, there is also a debate about the actual function of Nlgns in regulating basal excitatory synaptic transmission and plasticity. Several studies relying on the expression of Nlgn mutants have revealed the potential for Nlgn1 to recruit both NMDA receptors (NMDARs) and AMPA receptors (AMPARs) at synapses through extracellular and intracellular interactions, respectively (*Budreck et al., 2013*; *Giannone et al., 2013*; *Haas et al.,*

*2018*; *Heine et al., 2008*; *Letellier et al., 2018*; *Mondin et al., 2011*; *Shipman and Nicoll, 2012*). However, constitutive or conditional Nlgn1/2/3 KO selectively affect basal NMDAR-mediated EPSCs and not AMPAR-mediated EPSCs, and rescue experiments with truncated Nlgn1 mutants suggest that the synaptic recruitment of NMDARs requires the intracellular domain of Nlgn1 (*Chanda et al., 2017*; *Chubykin et al., 2007*; *Jiang et al., 2017*; *Wu et al., 2019*). Finally, while it is generally accepted that NMDAR-dependent long-term potentiation (LTP) is impaired by Nlgn1 KD or KO, the issues of which Nlgn1 motifs are important in this process and whether the Nlgn1-NMDAR interaction is required, are unclear (*Jiang et al., 2017*; *Kim et al., 2008*; *Letellier et al., 2018*; *Shipman and Nicoll, 2012*; *Wu et al., 2019*).

In addition to differences in experimental preparations, these studies relying on the manipulation of the Nlgn expression level all have potential biases, including the compensatory expression of proteins in the case of KO (*Dang et al., 2018*), off-target effects of inhibitory RNAs (*Alvarez et al., 2006*), and mislocalization of overexpressed Nlgns, for example Nlgn1 at inhibitory synapses and Nlgn2 at excitatory synapses (*Chih et al., 2006*; *Letellier et al., 2018*; *Nguyen et al., 2016*; *Tsetsenis et al., 2014*). Furthermore, these techniques operate on a long-term basis, that is days to weeks, due to slow protein turnover. Hence, there is a pressing need for alternative paradigms allowing for an acute control of neuroligin signaling pathways (*Jeong et al., 2017*) without affecting its expression level. Optogenetics is ideally suited for such purpose and was successfully implemented not only to regulate neuronal excitability and homeostasis, but also for fine tuning of protein-protein interactions and signaling pathways in neurons with light (*Berlin and Isacoff, 2017*; *Chang et al., 2014*; *Goold and Nicoll, 2010*; *Grubb and Burrone, 2010*; *Mao et al., 2018*; *Schwechter et al., 2013*; *Sinnen et al., 2017*; *Zhang et al., 2011*).

To acutely control Nlgn1 activity, we manipulated the phosphotyrosine level of endogenous Nlgn1 using a photoactivatable receptor tyrosine kinase targeting a unique intracellular tyrosine in Nlgn1 (Y782). This residue belongs to the gephyrin-binding motif and previous experiments showed that unphosphorylated Y782 strongly binds gephyrin - as does a Y782F mutant - while phosphorylated Y782 weakly binds gephyrin, a behavior phenocopied by a Y782A mutant (*Giannone et al., 2013*; *Letellier et al., 2018*). In parallel, neuronal expression of Nlgn1 Y782A (but not Y782F) promotes dendritic spine density and recruitment of PSD-95 and AMPARs (*Giannone et al., 2013*; *Letellier et al., 2018*), suggesting that Nlgn1 tyrosine phosphorylation is responsible for these effects. Here, we report that the stimulation of a light-gated version of the fibroblast growth factor receptor 1 (FGFR1) expressed in hippocampal CA1 neurons increases dendritic spine number as well as AMPAR-receptor-mediated EPSCs, and partially blocks LTP, in a Nlgn1 selective fashion, thus demonstrating a major role of the intracellular tyrosine phosphorylation of endogenous Nlgn1 in post-synaptic differentiation. Together, our results show that not only Nlgn1 is important for regulating dendritic spine number, but also that the Nlgn1 intracellular domain mediates AMPAR recruitment in basal conditions and regulates LTP.

## Results

### Light stimulation of Nlgn1 tyrosine phosphorylation

Using an in vitro kinase assay on recombinant GST fused to the intracellular domain of Nlgn1, we previously identified several tyrosine kinases able to directly phosphorylate Nlgn1, including Trk family receptors and the FGFR1 (*Letellier et al., 2018*). To acutely control Nlgn1 phosphorylation independently of endogenous ligand-activated kinases, we thus used here a photoactivatable version of FGFR1 (optoFGFR1) (*Grusch et al., 2014*; *Figure 1A*). To show that Nlgn1 can be acutely phosphorylated by optoFGFR1 in a light-dependent manner, we illuminated COS-7 cells co-expressing recombinant Nlgn1 and optoFGFR1 at 470 nm for 15 min using a light emitting diode (LED) array (*Figure 1B*). The stimulation of optoFGFR1 by light induced as much Nlgn1 phosphorylation as constitutively active FGFR1 (*Figure 1C,D*) (conditions CA and opto+, respectively), indicating potent kinase activation, while samples kept in the dark (conditions light-) did not show significant pTyr levels, revealing no unspecific effect of light. Finally, no phosphorylation of the point mutant Nlgn1-Y782F was observed upon light application (*Figure 1C,D*), demonstrating that Y782 is the only tyrosine residue on Nlgn1 which is phosphorylated by light-gated optoFGFR1.

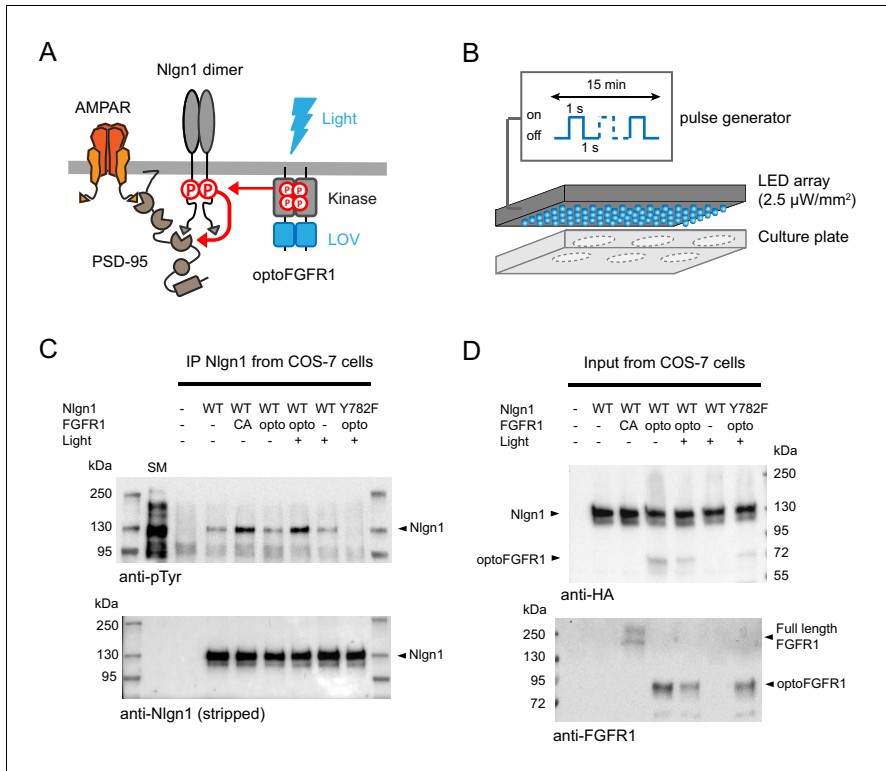

**Figure 1.** Optogenetic phosphorylation of Nlgn1 at residue Y782. (**A**) Schematic diagram of optogenetically-driven Nlgn1 tyrosine phosphorylation using optoFGFR1. Phosphorylated Nlgn1 is expected to recruit PSD-95 that serves as a platform for trapping AMPARs. (**B**) Scheme representing the 470 LED array that is placed in the incubator and used to illuminate COS-7 cells or organotypic slices contained in a six-well plate. (**C**) pTyr and Nlgn1 immunoblots of proteins extracted from COS-7 cells and immunoprecipitated with anti-Nlgn1 antibodies. Cells expressed either no Nlgn1, Nlgn1 alone, Nlgn1 + constitutively active (CA) FGFR1, Nlgn1 + optoFGFR1, and Nlgn1-Y782F + optoFGFR1. In the first lane, the starting material (SM) from non-transfected cells reveals numerous tyrosine phosphorylated proteins, whereas a single band is present in the Nlgn1 IP samples (black arrowhead). Cells were either kept in the dark (- light), or exposed to alternative 470 nm light and pulses (1 s light pulse every 1 s) for 15 min (+ light). (**D**) Corresponding starting material immunoblotted with HA and FGFR1 antibodies, respectively. The arrowheads represent HA-tagged Nlgn1, HA-tagged optoFGFR1, or constitutively active FGFR1.

## Light activation of optoFGFR1 increases dendritic spine density

We then examined the impact of triggering Nlgn1 tyrosine phosphorylation on synapse morphology and function in mouse organotypic hippocampal cultures, using confocal microscopy and electrophysiology, respectively (*Figure 2A*). Using single-cell electroporation, we expressed optoFGFR1 with a tdTomato volume marker in CA1 neurons of hippocampal slices obtained from either wild type or Nlgn1 KO mice. Immunostained HA-tagged optoFGFR1 was detected throughout dendrites including spines, that is at the right location to phosphorylate Nlgn1 (*Figure 2B*). Dendritic spine density increased by ~25% in neurons exposed to 470 nm light pulses for 24 hr, but remained stable in neurons expressing optoFGFR1 and kept in the dark, or in light-stimulated CA1 neurons from Nlgn1 KO slices (*Figure 2C,D*), demonstrating that this effect is mediated by light-dependent tyrosine phosphorylation of endogenous Nlgn1.

## Light activation of optoFGFR1 enhances basal AMPAR-, but not NMDAR-mediated EPSCs

At the electrophysiological level, we measured both AMPAR- and NMDAR-mediated EPSCs evoked by the stimulation of Schaffer's collaterals, comparing neurons expressing optoFGFR1 with paired non-electroporated neighbors by dual patch-clamp recordings (*Figure 3A*). Strikingly, neurons expressing optoFGFR1 and exposed to light for 24 hr exhibited ~200% larger evoked AMPAR-

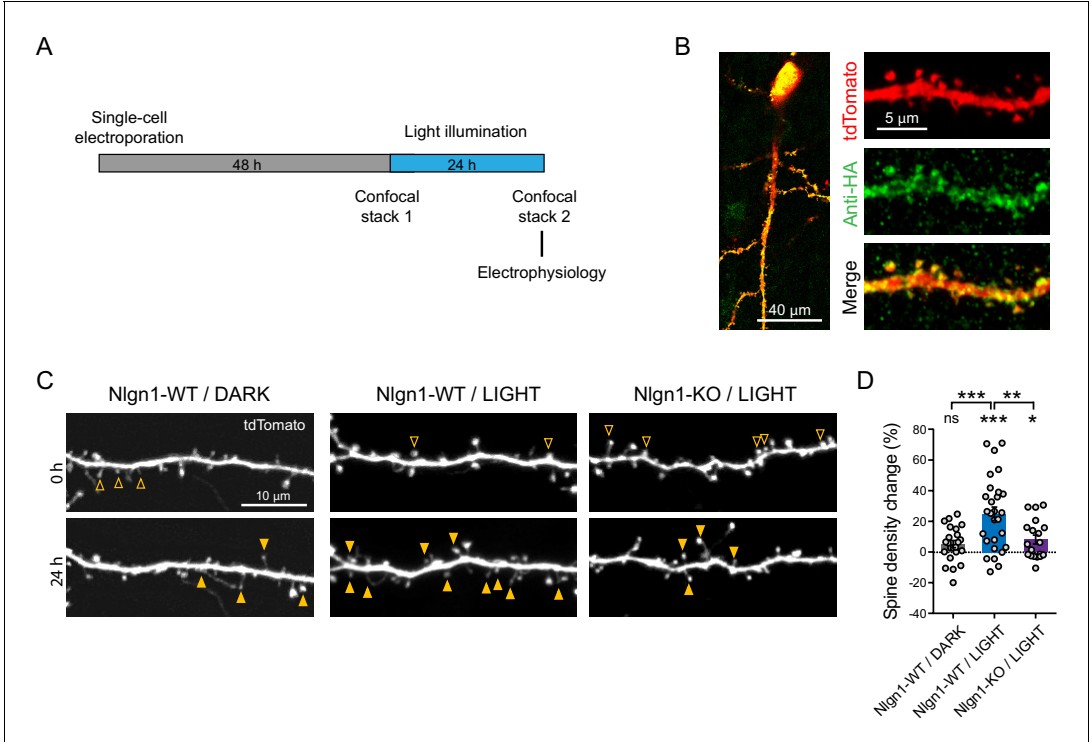

**Figure 2.** Optogenetic stimulation of optoFGFR1 increases dendritic spine density in a Nlgn1-dependent manner. (**A**) Experimental procedure to investigate the effect of the optogenetic stimulation of optoFGFR1 on spine density and synaptic transmission. CA1 neurons in organotypic slices from WT or Nlgn1 KO mice were electroporated at DIV 3–5 with tdTomato and HA-tagged optoFGFR1. Two days later, they were either stimulated with alternating blue light for 24 hr or kept in the dark, and processed for imaging or electrophysiology. (**B**) Confocal images of CA1 neurons and dendritic segments showing tdTomato (red) and anti-HA immunostaining (green). (**C**) Confocal images of apical dendrites from electroporated neurons before (0 hr) and 24 hr after light activation of optoFGFR1. Control slices did not receive light, or received light but were from the Nlgn1 KO background. Solid arrowheads point to spines which have appeared, and empty arrowheads to spines which have disappeared in the time interval. (**D**) Normalized spine density for each condition (n = 19–28 dendrites from N = 5–7 cells). Change in spine density was assessed for each condition using paired t-test (***p<0.001, *p<0.05, ns: not significant). Spine density change was compared across conditions using a one-way ANOVA followed by Tukey's multiple comparison test (***p<0.001, **p<0.01).

mediated EPSCs compared to non-electroporated neighbors that also received light, or to neurons expressing optoFGFR1 and kept in the dark (*Figure 3D,F*). This was accompanied by an almost two-fold increase in the frequency of spontaneous AMPAR-mediated EPSCs (*Figure 3B,C*), in agreement with the higher number of dendritic spines. No change in the amplitude or kinetics of spontaneous AMPAR-mediated EPSCs was measured (*Figure 3—figure supplement 1A–C*), indicating that optoFGFR1 activation did not change AMPAR channel conductance. In parallel, there was no significant impact of optoFGFR1 expression and/or light on evoked NMDAR-mediated EPSCs (*Figure 3E, G*). Importantly, the light-induced increase AMPAR-mediated EPSCs was not observed in CA1 neurons from Nlgn1 KO slices (*Figure 3D,F*), demonstrating that this effect involves the selective tyrosine phosphorylation of Nlgn1. The paired-pulse ratio was not changed by optoFGFR1 expression or light exposure, suggesting that presynaptic function was unaltered (*Figure 3—figure supplement 1D,E*). Furthermore, although the critical tyrosine residue belonging to the gephyrin-binding motif is conserved in Nlgn2 and Nlgn3 (*Poulopoulos et al., 2009*), where it can also be phosphorylated (*Letellier et al., 2018*), no effect of optoFGFR1 stimulation was observed on inhibitory currents recorded in CA1 neurons (*Figure 3—figure supplement 1F,G*). Together, these data demonstrate that the phosphorylation mechanism is specific to the Nlgn1 isoform at excitatory post-synapses, and selectively affects AMPAR recruitment at pre-existing or newly formed spines.

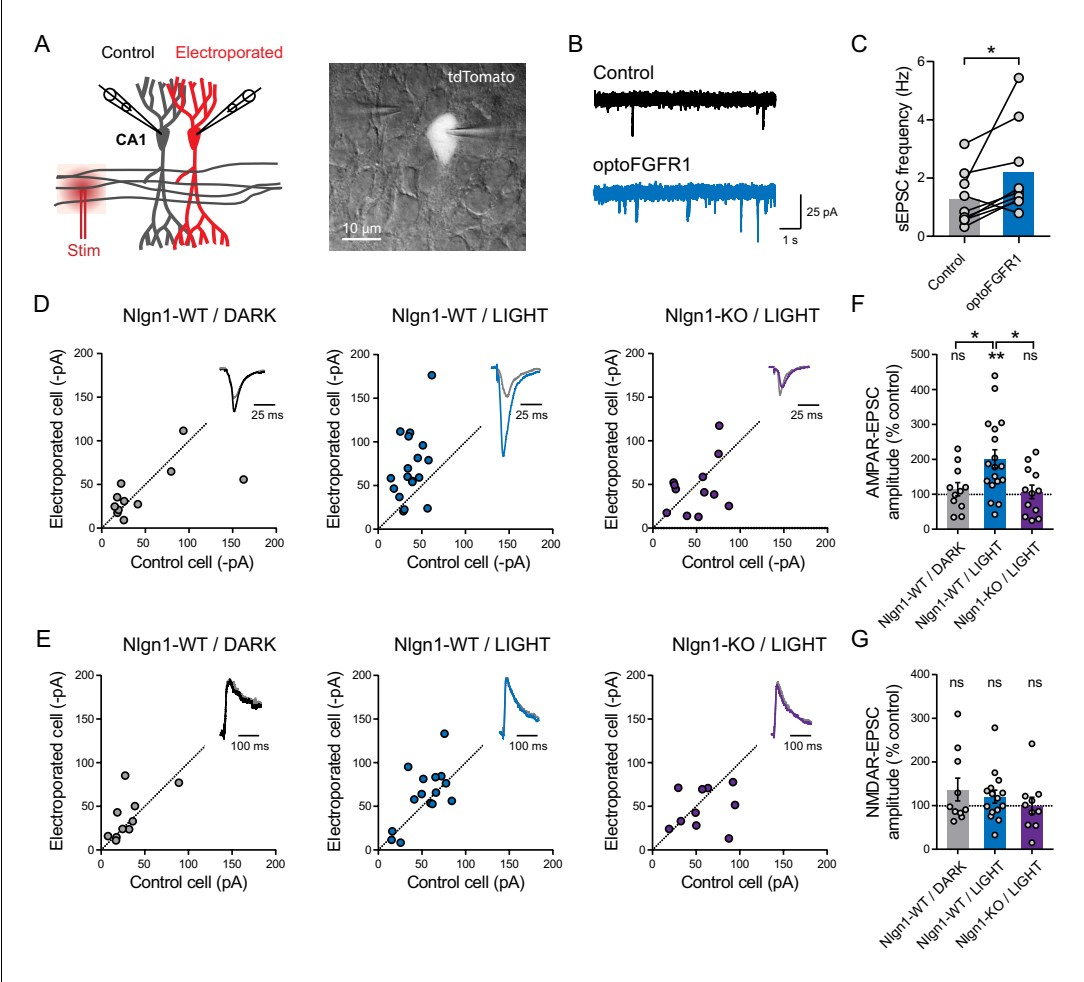

**Figure 3.** Light activation of optoFGFR1 in CA1 neurons selectively increases AMPA receptor-mediated EPSCs in a Nlgn1-dependent manner. (**A**) Dual patch-clamp recordings of AMPAR- and NMDAR-mediated currents upon stimulation of Schaffer's collaterals were made at holding potentials of −70 and +40 mV (respectively) in electroporated neurons and non-electroporated counterparts. The image shows two patched neurons in the CA1 area, the brighter one expressing optoFGFR1 + tdTomato. (**B**) Representative electrophysiological recordings of spontaneous AMPAR-mediated EPSCs (sEPSCs) in a control non-electroporated neuron (black trace), and a paired neuron expressing optoFGFR1 and pre-exposed to light for 24 hr (blue trace). (**C**) Corresponding sEPSC frequency for control neurons and neurons expressing optoFGFR1. Data were compared using the Wilcoxon matched-pairs signed rank test (*p<0.05). (**D, E**) Scatter plots of AMPAR- and NMDAR-mediated EPSCs, respectively, in neurons expressing optoFGFR1 compared to paired unelectroporated neurons (control cell), in the indicated conditions. Representative traces (color) normalized to control (grey) are shown as insets. (**F, G**) Average of AMPAR- and NMDAR-mediated EPSCs in the three conditions, normalized to the control (100%). Data were compared to the control condition by the Wilcoxon matched-pairs signed rank test, and between themselves using one-way ANOVA followed by Tukey's multiple comparison (**p<0.01, *p<0.05, ns: not significant).

The online version of this article includes the following figure supplement(s) for figure 3:

**Figure supplement 1.** Stimulation of optoFGFR1 does not affect spontaneous AMPAR-mediated EPSC amplitude or kinetics, paired pulse ratio (PPR), or inhibitory currents in CA1 neurons.

## The intracellular Y782 residue is involved in light-induced effects

To verify that optoFGFR1 was specifically phosphorylating the Nlgn1 Y782 residue in neurons, we adopted a replacement strategy (*Letellier et al., 2018*) by co-electroporating opto*Fgfr1*, *Nlgn1*-shRNA, and *Nlgn1* rescue constructs in slices from WT mice (*Figure 4A,B*). This led to basal AMPAR- and NMDAR-mediated EPSCs in the dark matching those measured in paired non-electroporated neurons expressing endogenous Nlgn1 levels (*Figure 4C–F*). In parallel, the density of dendritic spines remained stable over time in neurons expressing optoFGFR1 and kept in the dark, or not expressing optoFGFR1 and exposed to light, indicating no side effects

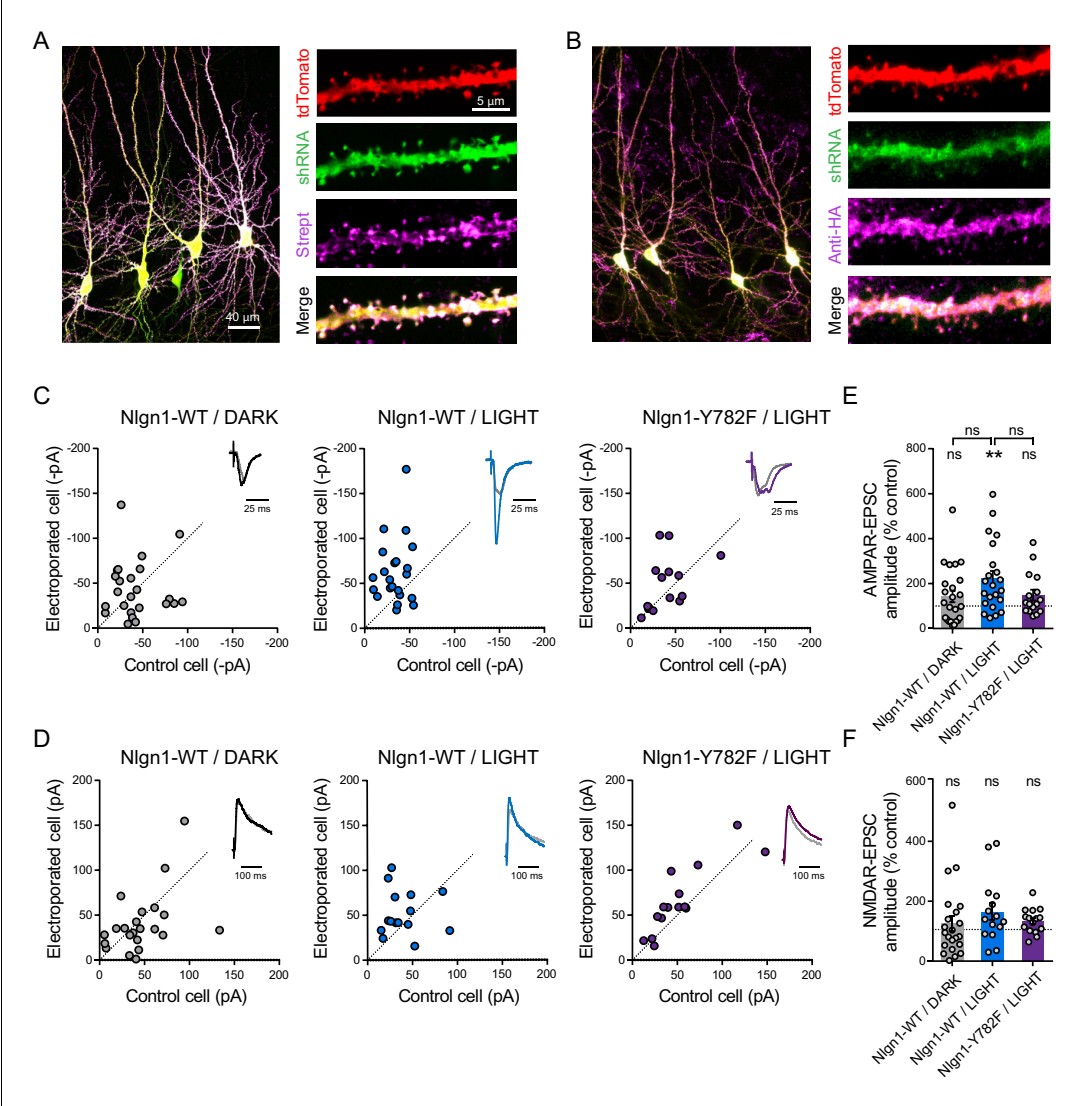

**Figure 4.** The light-induced increase in AMPA receptor-mediated EPSCs is specific to Y782 phosphorylation in Nlgn1. CA1 Neurons were co-electroporated with plasmids encoding tdTomato, shRNA against endogenous *Nlgn1* containing a GFP reporter, shRNA resistant AP-tagged *Nlgn1*-WT or -Y782F, biotin ligase (BirAER), and HA-tagged opto*Fgfr1*. (A, B) Confocal images showing tdTomato (red) and GFP (green). Biotinylated Nlgn1 and optoFGFR1 were stained in different slices using streptavidin-Atto647 and anti-HA antibody, respectively (magenta). (C, D) Scatter plots of AMPAR- and NMDAR-mediated EPSCs, respectively, for electroporated versus paired non-electroporated neurons (control cell) in the indicated conditions. Representative traces (black, blue or violet) normalized to control (grey) are shown as insets. (E, F) Average of AMPAR-and NMDAR-mediated EPSCs, respectively, normalized to the control (100%) in the different conditions. Data were compared to the control condition by the Wilcoxon matched-pairs signed rank test and between themselves using one-way ANOVA followed by Tukey's multiple comparison (**p<0.01, ns: not significant).

The online version of this article includes the following figure supplement(s) for figure 4:

**Figure supplement 1.** The light-induced increase in spine density is specific to Y782 phosphorylation in Nlgn1.

of either optoFGFR1 electroporation or photo-stimulation (*Figure 4—figure supplement 1*). In CA1 neurons expressing rescue Nlgn1-WT and optoFGFR1, light exposure induced again a ~25% increase in dendritic spine number (*Figure 4—figure supplement 1*), as well as a ~200% increase in AMPAR (but not NMDAR) -mediated EPSCs compared to control non-electroporated neurons (*Figure 4C–F*). These effects were similar to those found in neurons expressing endogenous Nlgn1 (*Figures 2* and *3*), validating the Nlgn1 replacement strategy. Importantly, the increase in spine density and AMPAR-mediated EPSCs by optoFGFR1 activation was not

observed in CA1 neurons expressing Nlgn1-Y782F (*Figure 4C,E* and *Figure 4—figure supplement 1*), indicating that those effects are mediated by phosphorylation of the Nlgn1 Y782 residue induced by the photo-activation of optoFGFR1.

## Light activation of Nlgn1 tyrosine phosphorylation impairs LTP

Finally, we asked whether the increase of basal AMPAR-mediated currents induced by Nlgn1 phosphorylation could partially occlude NMDAR-dependent long term potentiation (LTP). Using a pairing protocol, we induced an increase of about threefold in evoked AMPAR-mediated EPSCs, which was blocked by the NMDAR antagonist AP5 (*Figure 5—figure supplement 1A*). CA1 neurons expressing optoFGFR1 and pre-exposed to light showed a significant ~50% reduction in the LTP plateau level compared to control non-electroporated neighbors (*Figure 5A,B*). To check if this effect was again specific of Nlgn1, we performed LTP experiments in hippocampal slices from Nlgn1 KO mice.

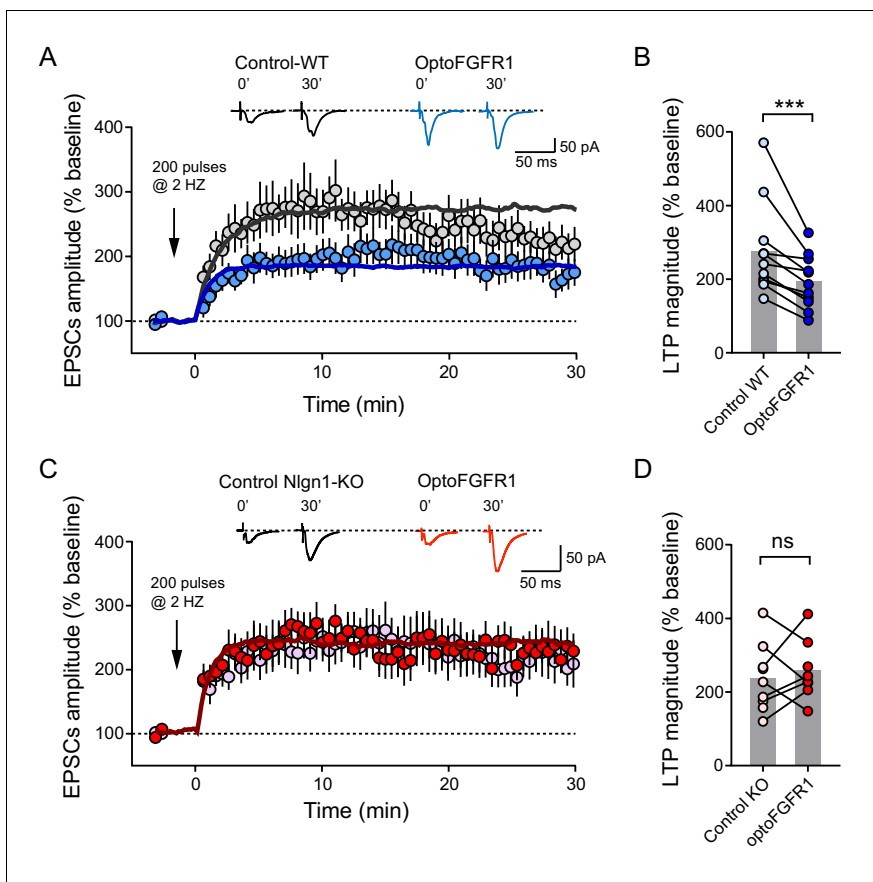

**Figure 5.** Light activation of Nlgn1 phosphorylation by optoFGFR1 reduces LTP. (**A**) Average AMPAR-mediated EPSCs in CA1 neurons expressing optoFGFR1 (blue circles) or in non-electroporated neighbors (grey circles), all pre-exposed to light for 24 hr, upon LTP induction at time 0 using a pairing protocol. Sample traces are shown at time 0 and 30 min after LTP induction. The solid lines show averages from 10 computer simulations for each condition with parameters $k_{on}$ = 1 s$^{-1}$ (black) or 3 s$^{-1}$ (blue), and $k_{off}$ drops from 0.04 to 0.004 s$^{-1}$. (**B**) Individual values of the long-term plateau of AMPAR-mediated EPSC in the two conditions (6–10 min after LTP induction), expressed as a percentage of the baseline level. Data were compared to the control condition (unelectroporated neuron also exposed to light) by Wilcoxon matched-pairs signed rank test (\*\*\*p<0.001). (**C, D**) Similar LTP recordings and quantification in CA1 neurons from Nlgn1 KO slices, with corresponding simulations (red line, $k_{on}$ = 1 s$^{-1}$ and $k_{off}$ drops from 0.04 to 0.008 s$^{-1}$). Data were compared to the control condition (unelectroporated neuron also exposed to light) by Wilcoxon matched-pairs signed rank test (ns: non-significant).

The online version of this article includes the following figure supplement(s) for figure 5:

**Figure supplement 1.** LTP dependence on NMDARs and effect of Nlgn1 expression level on the NMDA/AMPA ratio.

Surprisingly, the LTP level was barely reduced (*Figure 5A,C*), despite a significant decrease in NMDA/AMPA ratio in control non-electroporated Nlgn1 KO neurons compared to neurons from WT mice (*Figure 5—figure supplement 1B*; *Budreck et al., 2013*; *Chubykin et al., 2007*). However, light stimulation of optoFGFR1 did not alter LTP in Nlgn1 KO neurons (*Figure 5C,D*), indicating that Nlgn1 phosphorylation is responsible for the decreased LTP in wild-type neurons.

## Quantitative interpretation of LTP data by modeling AMPAR trapping at synapses

To quantitatively interpret those LTP results, we carried out computer simulations describing membrane diffusion and synaptic trapping of individual AMPARs (*Figure 6A*, *Figure 6—figure supplement 1A,B* and Supplementary source code), based on a previous framework using realistic kinetic parameters (*Czöndör et al., 2012*). This model is in line with experiments showing that hippocampal

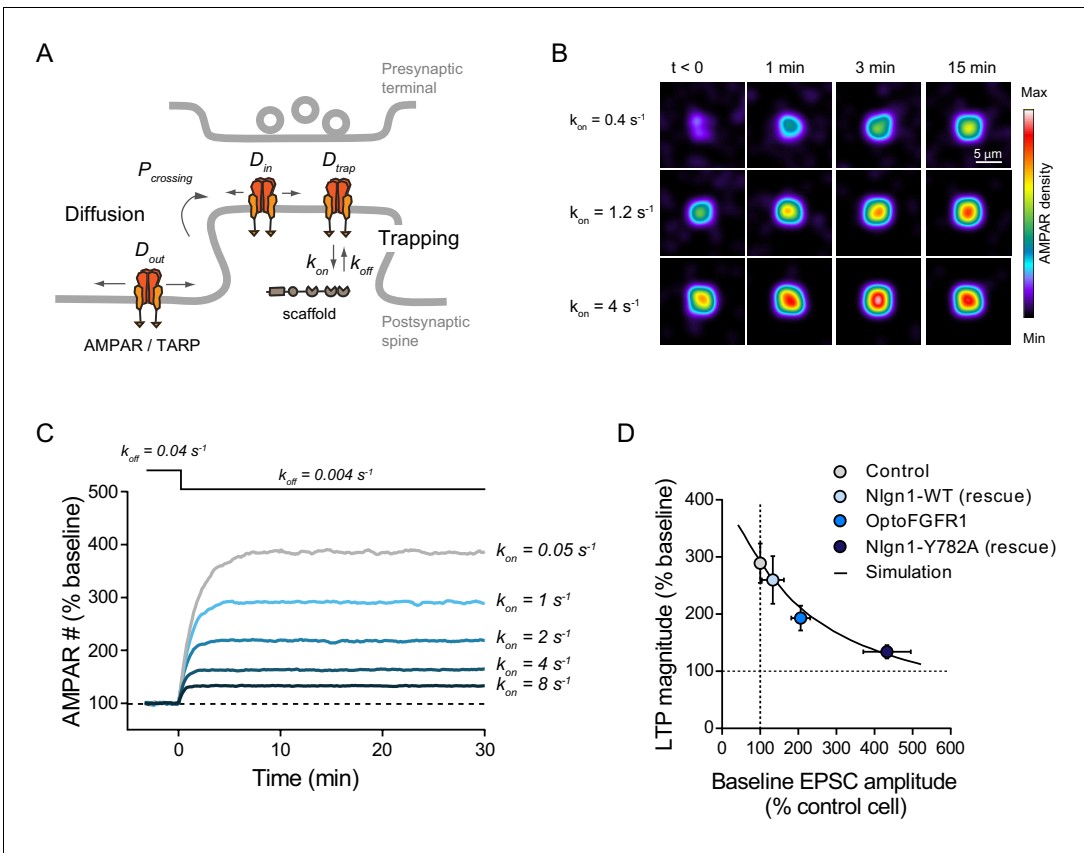

**Figure 6.** Modeling LTP experiments. (A) Schematic diagram of the AMPAR diffusion/trap process, with model parameters being indicated in italic. (B) Heat maps representing simulations of AMPAR accumulation at a single synapse over time. LTP is mimicked by decreasing the AMPAR/scaffold off-rate, causing a diffusional trapping of AMPARs from extra-synaptic pools. Nlgn1 phosphorylation is mimicked by increasing the initial AMPAR/scaffold on-rate, resulting in higher basal AMPAR level and lower LTP (relatively to baseline). (C) Graph showing the relative change in synaptic AMPAR content (in % of baseline) over time, when LTP is simulated by a drop in $k_{off}$ from 0.04 to 0.004 s$^{-1}$ at time zero. Curves correspond to different values of the parameter $k_{on}$ ranging from 0.05 to 8 s$^{-1}$. (D) Relationship between basal synaptic AMPAR content and LTP plateau level (% of baseline). Experimental points (circles) were obtained from non-electroporated neurons (grey), neurons expressing opoFGFR1 and stimulated with light (blue), or neurons co-expressing shRNA to Nlgn1 and either Nlgn1-WT (light blue) or Nlgn1-Y782A (dark blue) data taken from *Letellier et al. (2018)*. Basal AMPAR-mediated EPSCs were normalized to match a synaptic level of 33 AMPARs in the control condition (*Levet et al., 2015*). The solid line represents simulated data generated by varying the AMPAR/scaffold binding rate from 0.1 to 10 s$^{-1}$.

The online version of this article includes the following figure supplement(s) for figure 6:

**Figure supplement 1.** Computer simulations of AMPAR diffusion/trapping in LTP.

LTP primarily involves the capture of extra-synaptic AMPARs (*Granger et al., 2013*; *Penn et al., 2017*). We mimicked LTP by introducing a step decrease in the apparent off-rate between AMPARs and the PSD scaffold (*Figure 6B,C*). The simulations matched very well experimental LTP (*Figure 5A,C*), both in terms of kinetics and plateau value (~270%), supporting this diffusion/trap model. To mimic the effect of Nlgn1 phosphorylation on postsynaptic density (PSD) assembly and AMPAR recruitment (*Letellier et al., 2018*), we raised the AMPAR/scaffold binding rate, resulting in a ~2 fold increase of basal synaptic AMPAR number (*Figure 6—figure supplement 1C*) reproducing the experimental data (*Figure 3D,F*). In response to the same LTP simulation, the relative increase in AMPAR number now reached only ~190%, as in optoFGFR1 experiments (*Figure 5A*). Thus, the partial occlusion of LTP observed upon optoFGFR1 stimulation can be explained by a high initial recruitment of synaptic AMPARs, which depletes the extra-synaptic AMPAR reservoir necessary for LTP. Accordingly, we previously reported an almost complete occlusion of LTP upon replacement of endogenous Nlgn1 by a Y782A mutant which phenocopies maximally phosphorylated Nlgn1 and increases AMPAR-mediated EPSCs by ~4 fold (*Giannone et al., 2013*; *Letellier et al., 2018*). Overall, our model predicts a negative correlation between basal synaptic AMPAR number and the ability to respond to LTP (*Figure 6—figure supplement 1D*), that perfectly fits the experiments (*Figure 6D*). These data suggest that Nlgn1 tyrosine phosphorylation impairs LTP by promoting high initial synaptic AMPAR levels.

## Discussion

Orthogonal to the traditional paradigms used to manipulate Nlgn expression level or replace Nlgn isoforms with truncated or mutated versions, this novel optogenetic approach allows for a fine tuning of the tyrosine phosphorylation of endogenous Nlgn1, revealing a strong role of Nlgn1 intracellular signaling in excitatory post-synapse differentiation. Our results show that Nlgn1 tyrosine phosphorylation specifically regulates dendritic spine number, mediates AMPAR recruitment in basal conditions, and impairs LTP.

Together, our results support a mechanism by which, in its tyrosine phosphorylated state, Nlgn1 preferentially recruits intracellular PDZ domain containing scaffolding proteins including PSD-95 (*Giannone et al., 2013*; *Jeong et al., 2019*), associated with a morphological stabilization of dendritic spines (*Cane et al., 2014*) and serving as slots for the diffusional trapping of surface AMPARs (*Czöndör et al., 2013*; *Haas et al., 2018*; *Mondin et al., 2011*). In contrast, NMDAR-mediated EPSCs are not affected by Nlgn1 phosphorylation, supporting the concept of a direct extracellular coupling between Nlgn1 and GluN1 (*Budreck et al., 2013*; *Shipman and Nicoll, 2012*). The selective effects of endogenous Nlgn1 tyrosine phosphorylation on dendritic spine density, AMPAR-receptor-mediated EPSCs, and LTP are in close agreement with our previous observations based on the KD plus rescue of Nlgn1 point mutants, in particular the Nlgn1 Y782A mutant, which promotes synaptic recruitment of PSD-95, strongly enhances basal AMPAR-mediated EPSCs, and totally blocks LTP through synapse unsilencing mechanisms (*Letellier et al., 2018*). The increase in frequency - but not amplitude - of spontaneous AMPAR-mediated EPSCs upon optoFGFR1 stimulation indicates the formation of synapses containing a fixed bolus of AMPARs. Importantly, optogenetic Nlgn1 phosphorylation induces a similar response as PSD-95 overexpression by increasing spine density, enhancing AMPAR- but not NMDAR-dependent transmission, and occluding LTP (*Ehrlich and Malinow, 2004*; *El-Husseini et al., 2000*; *Stein et al., 2003*), which further supports our model. Conversely, unphosphorylated Nlgn1 (as mimicked by the Y782F mutant) might associate instead with gephyrin clusters shown to dynamically appear and disappear on the sides of dendritic spines (*Villa et al., 2016*). Thus, by locally controlling Nlgn1 phosphorylation, a single spine might have the possibility to assemble either excitatory or inhibitory scaffolding nanomodules (*Haas et al., 2018*; *Hruska et al., 2018*; *Tang et al., 2016*).

Our computer simulations of AMPAR diffusional trapping at PSDs provide a simple framework to interpret the experimental data. By avoiding gephyrin binding and instead triggering PSD scaffold assembly, Nlgn1 tyrosine phosphorylation provides synapses with fresh surface-diffusing AMPARs. These already potentiated (or unsilenced) synapses are thus less prone to respond to the LTP stimulation, because extra-synaptic pools of AMPARs have been consequently depleted (*Granger et al., 2013*; *Penn et al., 2017*). A strong role of Nlgn1/PSD-95 interaction in synaptic function is also supported by a recent study showing that PKA-mediated phosphorylation of the S839 residue located

near the C-terminal PDZ domain binding motif dynamically regulates PSD-95 binding, affecting both dendritic spine number and AMPAR-mediated miniature EPSCs (*Jeong et al., 2019*).

Although the tyrosine residue in Nlgn1 belongs to a gephyrin-binding motif which is highly conserved among the other Nlgn isoforms and controls gephyrin binding (*Giannone et al., 2013*; *Poulopoulos et al., 2009*), optoFGFR1 stimulation did not affect evoked inhibitory currents, suggesting that the phosphorylation of Nlgn2 or Nlgn3 either did not occur, or did not modify the recruitment of the gephyrin scaffold and associated GABA$_A$ receptors. The lack of effects of optoFGFR1 stimulation in neurons from Nlgn1 KO slices confirms that it is indeed Nlgn1 phosphorylation which is causing the observed increases in dendritic spine density and AMPAR-mediated EPSCs. It might be interesting to apply similar optogenetic approaches to control the phosphorylation of other intracellular Nlgn1 residues including S839 and T709 by engineering photoactivatable versions of PKA and CamKII, respectively (*Bemben et al., 2014*; *Jeong et al., 2019*), or inhibitors of those kinases (*Murakoshi et al., 2011*), expecting to alter Nlgn1 trafficking and thereby synaptic function and potentiation. Other phosphorylation sites have been reported in Nlgn2 and Nlgn4 which might also be interesting to target with such light-gated kinases (*Antonelli et al., 2014*; *Bemben et al., 2015a*).

Because optoFGFR1 is lacking a ligand-binding domain, its light-activation is expected to by-pass the endogenous regulation of Nlgn1 tyrosine phosphorylation, which involves the Trk family of tyrosine kinases (*Letellier et al., 2018*) that are responding to intrinsic ligands (BDNF and NGF) (*Harward et al., 2016*). Photoactivatable versions of Trks have been reported and their stimulation with light for 48 hr induces neurite outgrowth in DIV1-3 dissociated neurons and de novo formation of axonal filopodia within 30 min (*Chang et al., 2014*), but the effects on spine formation and synaptic transmission in mature neurons have not been measured yet. Short-term photoactivation of another tyrosine kinase, EphB2, leads within seconds to the retraction of non-stabilized dendritic filopodia (*Mao et al., 2018*) and within minutes to the induction of new filopodia by activating actin polymerization (*Locke et al., 2017*). Those effects are likely to obey different downstream signaling pathways than the ones we report here and which highly depend on Nlgn1 and the associated PSD scaffold.

Our data demonstrating the critical role of a single tyrosine residue located in the middle of the intracellular motif are difficult to reconcile with a previous report showing that a Nlgn3 construct with a 77-aa intracellular truncation (thus removing the motif containing the tyrosine) can still rescue AMPAR-mediated synaptic transmission upon Nlgn1/2/3 KD (*Shipman et al., 2011*). Moreover, whereas Nlgn1 KO was shown to affect primarily basal NMDAR-mediated synaptic transmission, we find instead strong effects of acute Nlgn1 tyrosine phosphorylation on basal AMPAR-mediated EPSCs, and no alteration of NMDAR-dependent EPSCs. The fact that AMPAR-mediated EPSCs are not altered in the Nlgn1 KO (*Chanda et al., 2017*) may result from the compensatory expression of scaffolding or adhesion molecules, in particular Nlgn3 (*Dang et al., 2018*), which also interacts with PSD-95. This would explain the fact that a dual Nlgn1/3 (and triple Nlgn1/2/3) KO are required to alter AMPARs levels and AMPAR-mEPSCs in cultured neurons (*Chanda et al., 2017*). In contrast, a compensatory expression of Nlgn3 which does not interact extracellularly with NMDARs (*Budreck et al., 2013*; *Shipman and Nicoll, 2012*) is not expected to rescue the decrease in NMDAR-EPSCs caused by Nlgn1 KO.

Finally, increasing the Nlgn1 phosphorylation by optoFGFR1 activation reduced LTP, as did the expression of the non-phosphorylatable Nlgn1-Y782F mutant (*Letellier et al., 2018*), suggesting that an optimal level of intracellular Nlgn1 tyrosine phosphorylation is necessary to elicit normal LTP. In contrast, another study found that LTP is impaired in acute slices from Nlgn1/2/3 cKO and can be rescued upon expression of a GPI-anchored Nlgn1 lacking the entire intracellular domain (*Wu et al., 2019*), and thus the C-terminal PDZ domain binding motif which we find important for anchoring AMPARs through PSD-95 (*Letellier et al., 2018*; *Mondin et al., 2011*). Moreover, we did not find a significant decrease of LTP in neurons from constitutive Nlgn1 KO, in contrast to previous reports (*Budreck et al., 2013*; *Jiang et al., 2017*; *Kim et al., 2008*; *Shipman and Nicoll, 2012*). While the differences might come from the use of different experimental preparations (acute vs organotypic slices), LTP stimulation protocols, and perturbation approaches (KD or KO, each with specific timing with respect to the synaptogenesis period), we believe that our approach allowing for an acute control of a signaling mechanism associated with endogenous Nlgn1, demonstrates a strong role of the Nlgn1 intracellular domain in synaptic function. Besides clarifying the role of Nlgn1 at excitatory

synapses, the optogenetic phosphorylation of Nlgn1 provides the exciting opportunity to control in time and space synaptic connectivity and function, and has therefore a great potential for investigating the causality between synaptic plasticity and learning processes as well as the possible contribution of Nlgns to neuropsychiatric behaviors (*Bourgeron, 2015*).

# Materials and methods

## Key resources table

| Reagent type (species) or resource | Designation | Source or reference | Identifiers | Additional information |
|---|---|---|---|---|
| Gene (*Mus musculus*) | *Nlgn1* | NCBI | NM_138666.4 | |
| Gene (*Mus musculus*) | *Fgfr1* | NCBI | NM_010206.3 | |
| Strain, strain background (*Mus musculus*) | C57Bl/6J | Charles River | RRID:IMSR_JAX:000664 | |
| Genetic reagent (*Mus musculus*) | Nlgn1-KO | *Varoqueaux et al., 2006* (PMID:16982420) | RRID:MGI:3688627 | N. Brose (MPI Goettingen) C57Bl/6J background |
| Cell line (*Simian*) | COS-7 | ATCC | RRID:CVCL_0224 | |
| Biological sample (*Mus muscumus*) | Organotypic slices (350 µm) | This paper *Stoppini et al. (1991)* (PMID:1715499) | | Prepared from P5-P8 animals. From C57Bl/6J WT or Nlgn1-KO mice |
| Transfected construct *Mus musculus* | HA-Nlgn1 | P. Scheiffele (Biozentrum, Basel) | | In COS cells with X-TremeGENE kit |
| Transfected construct *Mus musculus* | Fgfr1 V561M-FLAG (CA) | This paper | | Obtained with In-Fusion HD Cloning Kit using *Fgfr1-V561M-F* and *Fgfr1-V561M-R* primers on the Fgfr1-Flag plasmid. In COS cells |
| Transfected construct *Mus musculus* | optoFgfr1-HA | *Grusch et al., 2014* (PMID:24986882) | (RRID:Addgene_58745 | In COS cells with X-TremeGENE kit. In organotypic slices by single cell electroporation. |
| Transfected construct *Mus musculus* | HA-Nlgn1 Y782F | *Giannone et al., 2013* (PMID:23770246) | | In COS cells with X-TremeGENE kit. |
| Transfected construct (*Synthetic*) | tdTomato | R. Tsien (UC San Diego, CA) | | In organotypic slices by single cell electroporation. |
| Transfected construct (*M. musculus*) | Nlgn1 -shRNA (targeting Nlgn1) | *Chih et al., 2005* (PMID:15681343) | RRID:Addgene_59339 | Gift from P. Scheiffele In organotypic slices by single cell electroporation. |
| Transfected construct (*M. musculus*) | AP-Nlgn1 rescue (shRNA resistant) | *Chamma et al., 2016* (PMID:26979420) | | In organotypic slices by single cell electroporation. |
| Transfected construct (*M. musculus*) | AP-Nlgn1 Y782F rescue (shRNA resistant) | *Letellier et al., 2018* (PMID:30266896) | | In organotypic slices by single cell electroporation. |
| Transfected construct (*M. musculus*) | BirA$^{ER}$ | A. Ting (Stanford University, CA) | | |
| Recombinant DNA reagent (*M. musculus*) | *Fgfr1-Flag* plasmid | *Duchesne et al., 2006* PMID:16829530 | | L. Duchesne (Université de Rennes) |
| Sequence-based reagent | Primers *Fgfr1-V561M-F*; *Fgfr1-V561M-R* | This paper From Eurogentec | | *TGTCATTATGGA GTACGCCTC*; *TACTCCATAATGA CATAAAGAGG* |

*Continued on next page*

*Continued*

| Reagent type (species) or resource | Designation | Source or reference | Identifiers | Additional information |
|---|---|---|---|---|
| Antibody | anti-Nlgn1 (Rabbit polyclonal) | Synaptic systems 129013 | RRID:AB_2151646 | IP (2 µg) WB (1:1000) |
| Antibody | anti-phosphotyrosine P-Tyr-100 (Mouse monoclonal) | Cell Signaling Technology 9411 | RRID:AB_331228 | WB (1:1000) |
| Antibody | anti-FGFR1 (monoclonal polyclonal) | Cell Signaling Technology D8E4 9740 | RRID:AB_11178519 | WB (1:1000) |
| Antibody | anti-HA (Rat monoclonal) | Roche 3F10 11867423001 | RRID:AB_390918 | WB (1:1000) IHC (1:100) |
| Antibody | Easyblot HRP antibodies anti-mouse;anti-rabbit | GeneTex GTX221667-01; GTX221666-01 | RRID:AB_10728926; RRID:AB_10620421 | WB (1:1000) |
| Antibody | Alexa647-conjugated anti-rat antibody (Goat Polyclonal) | Molecular Probes A21247 | RRID:AB_141778 | IHC (1:200) |
| Peptide, recombinant protein | NeutrAvidin | Invitrogen | A2666 | IHC (1:200) |
| Commercial assay or kit | In-Fusion HD Cloning Kit | Takara Bio | 639642 (Ozyme) | In COS cells |
| Commercial assay or kit | X-tremeGENE HP DNA Transfection Reagent | Roche (RRID:SCR_001326) | 6366546001 | |
| Commercial assay or kit | Dynabeads Protein G | Thermo Fisher Scientific (RRID:SCR_008452) | 10004D | For immunoprecipitation |
| Chemical compound, drug | D-AP5 | TOCRIS (RRID:SCR_003689) | 0106/10 | 50 µM |
| Chemical compound, drug | Bicuculline | TOCRIS (RRID:SCR_003689) | 0130/50 | 20 µM |
| Chemical compound, drug | NBQX | TOCRIS (RRID:SCR_003689) | 0373/10 | 100 nM or 10 µM |
| Chemical compound, drug | NHS-ester ATTO 647N | ATTO-TEC GmbH | AD 647 N-31 | |
| Software, algorithm | Metamorph | Molecular Devices | RRID:SCR_002368 | |
| Software, algorithm | GraphPad | PRISM | RRID:SCR_002798 | |
| Software, algorithm | Clampex | Axon Instruments | | |
| Software, algorithm | Clampfit | Axon Instruments | | |
| Software, algorithm | Minianalysis | Synaptosoft | RRID:SCR_002184 | |
| Software, algorithm | FluoSim | *Lagardère et al., 2020* (DOI: 10.1101/2020.02.06.937045) | | Full source code will be deposited on GitHub upon paper acceptance |

## Constructs

Plasmids for BirA^ER and AP-*Nlgn1* harboring both extracellular splice inserts A and B were kind gifts from A. Ting (Stanford University, CA). Short hairpin RNA to murine *Nlgn1* (sh*Nlgn1*) was a generous gift from P. Scheiffele (Biozentrum, Basel). shRNA-resistant AP-tagged *Nlgn1* and *Nlgn1*-Y782F were described previously (*Chamma et al., 2016*; *Letellier et al., 2018*). The tdTomato plasmid was a generous gift from R. Tsien (UC San Diego, CA). *Fgfr1*-Flag (*Duchesne et al., 2006*) was a generous gift from L. Duchesne (Université de Rennes). To generate constitutively active (CA) *Fgfr1*-Flag, the V561M mutation was introduced using the In-Fusion HD Cloning Kit (Takara Bio) and the following primers: *Fgfr1*-V561M-F 5'TGTCATTATGGAGTACGCCTC3' and *Fgfr1*-V561M-R 5'TACTCCATAATGACATAAAGAGG3'. Opto*Fgfr1* bearing an

N-terminal myristoylation motif to attach to the membrane, and a C-terminal HA-tag was described previously (Grusch et al., 2014). In this construct, the extracellular FGF binding domain has been removed, and a light-oxygen voltage sensing (LOV) domain is fused to the C-terminus, such that stimulation with blue light induces dimerization of the FGFR1 intracellular domain and subsequent kinase activation in a ligand-independent manner.

## COS-7 cell culture and transfection

COS-7 cells purchased from ATCC (cell line source ECACC 87021302) were cultured in DMEM (Eurobio) supplemented with 2 mM glutamax (GIBCO), 1 mM sodium pyruvate (Sigma-Aldrich), and 10% Fetal Bovine Serum (Eurobio). COS-7 cells used in this study were regularly tested negative for Mycoplasma using the MycoAlertTM Detection Kit (LT07-218) from Lonza. For IP experiments, cells were plated in 6-well plates at a density of 120,000 per well. After 1 day, cells were transfected with 10:1 combinations of Nlgn1 and FGFR1 DNA constructs using the X-TremeGENE kit (Roche), with 1 μg total DNA for 2 μL reagent in 100 μL PBS per well. Cells were left under a humidified 5% $CO_2$ atmosphere (37°C) for 2 days before being processed for biochemistry.

## Organotypic slice culture

Organotypic hippocampal slice cultures were prepared as described (Stoppini et al., 1991) from either wild type or Nlgn1 knock-out mice (C57Bl/6J strain) obtained from N. Brose (MPI Goettingen). Animals were raised in our animal facility and were handled and killed according to European ethical rules. Briefly, animals at postnatal days 5–8 were quickly decapitated and brains placed in ice-cold Gey's balanced salt solution under sterile conditions. Hippocampi were dissected out and coronal slices (350 μm) were cut using a tissue chopper (McIlwain) and incubated at 35°C with serum-containing medium on Millicell culture inserts (CM, Millipore). The medium was replaced every 2–3 days.

## LED illumination

For cells expressing optoFGFR1, all steps before biochemical, confocal, or electrophysiological analysis were done in the dark. COS-7 cells or organotypic hippocampal slices were exposed to light pulses of 1 s every other second for 15 min or 24 hr, respectively, by placing the six-well plates under a custom-made rectangular array comprising 8 × 12 light emitting diodes (LEDs) (470 nm, 15 mW each), powered by a 24 V DC supply, and driven by an internal Arduino Leonardo pulse generator. The array was covered with a 5-mm-thick white Plexiglas sheet to dim the emitted light power by ~100 fold (2.5 μW/mm$^2$).

## Immuno-precipitation, SDS–PAGE, and immunoblotting

COS-7 cells were treated with 10 μM pervanadate for 15 min before lysis to preserve phosphate groups on Nlgn1. Whole-cell protein extracts were obtained by solubilizing cells in lysis buffer (50 mM HEPES, pH 7.2, 10 mM EDTA, 0.1% SDS, 1% NP-40, 0.5% DOC, 2 mM Na-Vanadate, 35 μM PAO, 48 mM Na-Pyrophosphate, 100 mM NaF, 30 mM phenyl-phosphate, 50 μM $NH_4$-molybdate and 1 mM $ZnCl_2$) containing protease Inhibitor Cocktail Set III, EDTA-Free (Calbiochem). Lysates were clarified by centrifugation at 8000 × g for 15 min. Equal amounts of protein (500 μg, estimated by Direct Detect assay, Merck Millipore) were incubated overnight with 2 μg rabbit anti-Nlgn1 (Synaptic systems 129013), then precipitated with protein G beads (Dynabeads Protein G, Thermo Fisher Scientific) and washed four times with lysis buffer. At the end of the immunoprecipitation, 20 μL beads were resuspended in 20 μL of 2x loading buffer (120 mM Tris-HCl, 3% SDS, 10% glycerol, 2% β-mercaptoethanol, 0.02% bromophenol blue, pH = 6.8). After magnetic beads isolation, half of the supernatants or starting materials (10–20 μg) were separated on 4–15% Mini-PROTEAN TGX Precast Protein Gels (Bio-Rad) and transferred to nitrocellulose membranes for immunoblotting (semi-dry, 7 min, Bio-Rad). After blocking with 5% non-fat dried milk in Tris-buffered saline Tween-20 (TBST; 28 mM Tris, 137 mM NaCl, 0.05% Tween-20, pH 7.4) for 45 min at room temperature, membranes were probed for 1 hr at room temperature or overnight at 4°C with mouse anti-phosphotyrosine (1:1000, Cell Signaling Technology 9411S), rabbit anti-Nlgn1 (1:1000, Synaptic systems 129013), rabbit anti-FGFR1 (1:1000, Cell Signaling Technology D8E4), or rat anti-HA (1:1000, Roche 3F10). After washing three times with TBST buffer, blots were incubated for 1 hr at room temperature with the

corresponding horseradish peroxidase (HRP)–conjugated goat secondary antibodies (1:5000, Jackson Immunoresearch) for input samples, or Easyblot HRP antibodies (GeneTex) for IP samples. The latter was used to avoid the detection of primary antibodies from the IP. Target proteins were detected by chemiluminescence with Super signal West Femto (Pierce) on the ChemiDoc Touch system (Bio-Rad).

### Single-cell electroporation

After 3–4 days in culture, organotypic slices were transferred to an artificial cerebrospinal fluid (ACSF) containing (in mM): 130 NaCl, 2.5 KCl, 2.2 $CaCl_2$, 1.5 $MgCl_2$, 10 D-glucose, 10 HEPES (pH 7.35, osmolarity adjusted to 300 mOsm). CA1 pyramidal cells were then processed for single-cell electroporation using glass micropipets containing plasmids encoding TdTomato (6 ng/μL) and optoFGFR1 (13 ng/μL). For rescue experiments, a plasmid carrying the Nlgn1 specific shRNA (13 ng/μL) was electroporated along with a resistant AP-Nlgn1 or Y782F mutant (13 ng/μL), $BirA^{ER}$ (6 ng/μL), TdTomato (6 ng/μL) and optoFGFR1 (13 ng/μL). Micropipets were pulled from 1 mm borosilicate capillaries (Harvard Apparatus) with a vertical puller (Narishige). Electroporation was performed by applying four square pulses of negative voltage (−2.5 V, 25 ms duration) at 1 Hz, then the pipet was gently removed. 10–20 neurons were electroporated per slice, and the slice was placed back in the incubator for 2–3 days before electrophysiology or confocal imaging.

### Immunohistochemistry

For visualization of recombinant AP-Nlgn1 and spine morphology in electroporated CA1 neurons expressing tdTomato, AP-Nlgn1 and $BirA^{ER}$, organotypic slices were fixed with 4% paraformaldehyde- 4% sucrose in PBS for 4 hr before the permeabilization of membranes with 0.25% Triton in PBS. Slices were subsequently incubated with NeutrAvidin (1:200, Invitrogen, A2226) conjugated to NHS-ester ATTO 647N (ATTO-TEC GmbH, AD 647 N-31) for 2 hr at room temperature. For visualization of HA-tagged optoFGFR1, fixed and permeabilized slices were incubated with rat anti-HA (Roche, clone 3F10, 1:100) overnight at 4°C. Slices were subsequently incubated with Alexa647-conjugated goat anti-rat antibody (Molecular Probes, 1:200) for 2 hr at room temperature.

### Confocal microscopy and spine counting

For fixed slices, images of single CA1 electroporated neurons co-expressing tdTomato, $BirA^{ER}$ and AP-Nlgn1 (WT or Y782F mutant) were acquired on a commercial Leica DMI6000 TCS SP5 microscope using a 63x/1.4 NA oil objective and a pinhole opened to one time the Airy disk. Images of 4096 × 4096 pixels, giving a pixel size of 70 nm, were acquired at a scanning frequency of 400 Hz. The number of optical sections was between 150–200, using a vertical step size of 0.3–0.4 μm. The number of spines per unit dendrite length of tdTomato-positive cells in secondary and tertiary apical dendrites was calculated manually using Metamorph (Molecular Devices).

To assess the effect of optoFGFR1 stimulation on the formation of dendritic spines, we took confocal stacks of the dendritic tree of several CA1 neurons before light stimulation, then exposed the organotypic slices to dim 470 nm light pulses (1 s pulse every 2 s, 2.5 μW/mm$^2$) through the LED array placed in the incubator for 24 hr, and finally took another round of images of the same neurons. For such time-lapse imaging, short imaging sessions (10–15 min) of live electroporated slices were carried out with a commercial Leica DMI6000 TCS SP5 microscope using a 63x/0.9 NA dipping objective and a pinhole opened to one time the Airy disk. Slices were maintained in HEPES-based ACSF. Laser intensity in all these experiments was kept at a minimum and acquisition conditions remained similar between the two imaging sessions. 12-bit images of 1024 × 1024 pixels, giving a pixel size of 120 nm, were acquired at a scanning frequency of 400 Hz. The number of optical sections varied between 150 and 200, and the vertical step size was 0.3–0.4 μm. The number of spines per unit dendrite length of tdTomato-positive neurons was calculated manually in Metamorph.

### Electrophysiological recordings

Whole-cell patch-clamp recordings were carried out at room temperature in CA1 neurons from organotypic hippocampal cultures, placed on a Nikon Eclipse FN1 upright microscope equipped with a motorized stage and two manipulators (Scientifica). CA1 pyramidal neurons were imaged

with DIC and electroporated neurons were identified by visualizing the GFP or Tdtomato fluorescence. The recording chamber was continuously perfused with ACSF bubbled with 95% $O_2$/5% $CO_2$ containing (in mM): 125 NaCl, 2.5 KCl, 26 $NaHCO_3$, 1.25 $NaH_2PO_4$, 2 $CaCl_2$, 1 $MgCl_2$, and 25 glucose. 20 µM bicuculline and 100 nM NBQX were added to block inhibitory synaptic transmission and reduce epileptiform activity, respectively. The series resistance Rs was left uncompensated, and recordings with Rs higher than 30 MΩ were discarded. We measured both AMPA- and NMDA-receptor mediated EPSCs upon electrical stimulation of Schaffer's collaterals, using a double-patch clamp configuration to normalize the recordings with respect to a neighboring non-electroporated neuron (*Shipman et al., 2011*). Voltage-clamp recordings were digitized using the Multiclamp 700B amplifier (Axon Instruments) and acquired using the Clampex software (Axon Instruments). EPSCs and IPSCs were evoked in an electroporated neuron and a nearby non-electroporated neuron (control) every 10 s for 5 min using a bipolar electrode in borosilicate theta glass filled with ACSF and placed in the stratum radiatum or pyramidal layer; respectively. AMPAR-mediated currents were recorded at −70 mV and NMDAR-mediated currents were recorded at +40 mV and measured 50 ms after the stimulus, when AMPAR-mediated EPSCs are back to baseline. IPSCs were recorded at +10 mV and in the presence of 10 µM NBQX and 50 µM D-AP5 to block AMPARs and NMDARs, respectively. EPSCs and IPSCs amplitude measurements were performed using Clampfit (Axon Instruments).

For LTP recordings, ACSF contained in (mM) 125 NaCl, 2.5 KCl, 26 $NaHCO_3$, 1.25 $NaH_2PO_4$, 4 $CaCl_2$, 4 $MgCl_2$, 25 glucose, and 0.02 bicuculline, while recording pipettes were filled with intracellular solution containing in mM: 125 Cs-MeSO$_4$, 10 CsCl, 10 HEPES, 2.5 $MgCl_2$, 4 $Na_2ATP$, 0.4 NaGTP and 10 phosphocreatine. Axons from CA3 pyramidal cells were cut with a scalpel to prevent spontaneous action potential propagation. Slices were maintained at 25°C throughout the recording. Baseline AMPAR-mediated EPSCs were recorded every 10 s for 2 min before LTP induction. Then LTP was induced by depolarizing the cells to 0 mV while stimulating the afferent Schaffer's collaterals at 2 Hz for 100 s. Recordings were sampled at −70 mV every 10 s for 30 min after LTP induction. In some recordings, LTP was induced in presence of 50 µm D-AP5 to block NMDARs. Between the stimulations, spontaneous AMPAR-mediated EPSCs (sEPSCs) were also recorded. sEPSC amplitude, frequency, rise time and decay times were measured from averaged sEPSCs using miniAnalysis (Synaptosoft).

## Computer simulations of AMPAR diffusion-trapping in LTP conditions

The computer program is based on a previous framework describing the role of AMPAR membrane dynamics in synaptic plasticity (*Czöndör et al., 2012*). Our original model included two types of processes to target AMPAR to synapses, that is diffusional trapping and vesicular recycling. However, based on recent experimental findings that hippocampal LTP primarily involves the diffusional trapping of extra-synaptic AMPARs (*Granger et al., 2013*; *Penn et al., 2017*), the current model focuses only on this process. Briefly, a dendritic segment is approximated by a 2D rectangular region (2 µm x 10 µm) containing five synapses (squares of 0.3 µm x 0.3 µm, surface area ~0.1 µm$^2$), corresponding to a linear density of 0.5 synapse/µm as measured experimentally (*Letellier et al., 2018*). This area is populated with 1000 AMPARs, initially placed at random positions. AMPARs are characterized by their 2D coordinates x and y, over time, *t*. When AMPARs reach the region contours, rebound conditions are applied to keep them inside, that is the system is closed. At each time step ($\Delta t$ = 100 ms), the coordinates are incremented by the distances $\Delta x = (2D\Delta t)^{1/2} n_x$ and $\Delta y = (2D\Delta t)^{1/2} n_y$, where $n_x$ and $n_y$ are random numbers generated from a normal distribution, and D is a diffusion coefficient which depends on whether AMPARs are outside ($D_{out}$ = 0.1 µm$^2$/s) or inside ($D_{in}$ = 0.05 µm$^2$/s) the synapse, values being taken from single molecule tracking experiments (*Nair et al., 2013*). Lower AMPAR diffusion within the synaptic cleft is attributed to steric hindrance. To introduce a diffusion barrier at the synapse (*Ewers et al., 2014*), AMPARs are allowed to cross the synaptic border with a probability $P_{crossing}$ = 0.5.

Within the synapse, AMPARs may reversibly bind to static post-synaptic density (PSD) components, namely PDZ domain containing scaffolding proteins including PSD-95, S-SCAM, PICK or GRIP, through the C-terminal PDZ motifs of GluA1/2, or of TARPs (*Bats et al., 2007*; *Kim and Sheng, 2004*). To describe those dynamic interactions, we define two global parameters, the AMPAR/scaffold binding and unbinding rates ($k_{on}$ = 1 s$^{-1}$ and $k_{off}$ = 0.04 s$^{-1}$, respectively), obtained by previously fitting SPT and FRAP experiments (*Czöndör et al., 2012*). AMPARs are

allowed to stay in the PSD if the probability of binding in this time interval ($k_{on}.\Delta t$) is greater than a random number between 0 and 1 generated from a uniform distribution. Otherwise, AMPARs continue to diffuse with coefficient $D_{in}$. When bound to the PSD, AMPARs move with a lower diffusion coefficient $D_{trap}$ = 0.006 µm$^2$/s, corresponding to confinement in the PSD (*Czöndör et al., 2013*; *Nair et al., 2013*). AMPARs stay in the PSD until their detachment probability ($k_{off}.\Delta t$), exceeds another random number. Then, AMPARs can bind the same PSD again or escape into the extra-synaptic space. At steady state (reached for $t > 1/k_{off}$), there is a dynamic equilibrium between synaptic and extra-synaptic AMPARs. The enrichment ratio between synaptic and extra-synaptic AMPAR density is given by the formula: $P_{crossing}$ ($D_{out}/D_{in}$) (1 + $k_{on}/k_{off}$). The maximal theoretical number of AMPARs per synapse is 200, when all extra-synaptic receptors in the system have been captured (given the excess of scaffolds versus AMPARs, we do not impose a saturation of binding sites here). With the chosen parameters however, there are about 30 AMPARs per synapse at basal state in control conditions, close to experimental measurements made by super-resolution imaging and freeze-fracture EM (*Levet et al., 2015*; *Shinohara et al., 2008*). The effect of Nlgn1 tyrosine phosphorylation on basal synaptic AMPAR levels was simulated by raising the AMPAR/scaffold binding rate ($k_{on}$), thereby mimicking an increase in the steady-state number of average post-synaptic AMPAR trapping slots observed experimentally (*Giannone et al., 2013*; *Letellier et al., 2018*; *Mondin et al., 2011*).

To simulate LTP, the AMPAR/scaffold unbinding rate ($k_{off}$) was decreased at time zero from higher (0.02 to 0.08 s$^{-1}$) to lower values (0.002–0.006 s$^{-1}$), hereby mimicking a higher affinity of TARPs to PSD-95 induced by CamKII activation (*Hafner et al., 2015*; *Opazo et al., 2010*). When we tried instead to simulate LTP by raising the parameter $k_{on}$ at time zero, the predicted time course was much more rapid than the one observed experimentally (i.e. the plateau was reached in about one minute). Thus, that type of mechanism is not likely to operate in the particular LTP protocol used here. The total length of the trajectories was set to 35 min, including a 5 min baseline, to match the whole duration of LTP experiments. Ten simulations were generated for each type of condition, and the number of AMPARs per synapse was determined and averaged (sem is within 1% of the mean). To determine the theoretical relationship between LTP level and basal synaptic AMPARs content, the parameter $k_{on}$ was varied between 0.075 s$^{-1}$ and 10 s$^{-1}$, thus simulating synapses that contain less or more AMPARs, respectively. We provide here as a supplemental file the original Mathematica source code described earlier to simulate LTP experiments (*Czöndör et al., 2012*). However, the algorithm used to make the simulations in this paper is part of a new, integrated software called FluoSim, which is submitted elsewhere (*Lagardère et al., 2020*) and whose source code will be made freely available through *github* once published.

## Sampling and statistics

For the analysis of dendritic spines observed by confocal microscopy, N et n values represent the total number of cells and dendrites, respectively. For each experiment, three to four independent dissections (from two to three animals) were used. Sample sizes were determined according to previous studies (*Letellier et al., 2018*; *Shipman et al., 2011*).

Summary statistics are presented as mean ± SEM (Standard Error of the Mean), including individual data points. Statistical significance tests were performed using GraphPad Prism software (San Diego, CA). Test for normality was performed with D'Agostino and Pearson omnibus normality test. Paired data obtained by imaging or electrophysiology experiments were compared using the Wilcoxon matched-pairs signed rank test when criteria for normality were not met. When paired data followed a normal distribution, we used a paired t-test. The non-electroporated neuron serves as a paired control, since it is patched simultaneously as the electoporated neuron and receives the same input fibers and stimulation. ANOVA test was used to compare means of several groups of normally distributes variables. Kruskal-Wallis test was used to compare several groups showing non-normal distributions. Dunn's multiple comparisons post hoc test was then used to determine the p value between two conditions. Statistical significance was assumed when $p < 0.05$. In the figures, *$p < 0.05$, **$p < 0.01$, ***$p < 0.001$, ****$p < 0.0001$.

## Ethical statement

The authors declare that they have complied with all relevant ethical regulations (study protocol approved by the Ethical Committee of Bordeaux CE50).

## Acknowledgements

We acknowledge L Duschene, P Scheiffele, and A Ting for the generous gift of plasmids, N Brose for the gift of Nlgn1 KO mice, A Hoagland for help with LED array construction and E Isacoff for access to laboratory resources, M Sainlos for fruitful discussions, the Bordeaux Imaging Center (C Poujol and S Marais) for support in microscopy, J Carrere, J Gautron, and R Sterling for technical assistance, the animal facility of the University of Bordeaux (in particular A Lacquemant S Pavelot, G Artaxet, G Dabee, E Normand, and C Martin), the cell culture facility of the Institute (especially M Munier, S Benquet, and E Verdier), the Biochemistry platform of the Neurocampus, and the Animal genotyping facility of NeuroCentre Magendie.

This work received funding from the Centre National de la Recherche Scientifique, Agence Nationale pour la Recherche (grant « Synthesyn » ANR-17-CE16-0028-01), Commission Franco-Américaine (Fulbright program), Conseil Régional Aquitaine (« SiMoDyn »), Investissements d'Avenir (Labex BRAIN ANR-10-LABX-43), Fondation pour la Recherche Médicale (« Equipe FRM » DEQ20160334916), and the national infrastructure France BioImaging (grant ANR-10INBS-04–01).

## Additional information

### Funding

| Funder | Grant reference number | Author |
| --- | --- | --- |
| Agence Nationale de la Recherche | ANR-17-CE16-0028-01 | Olivier Thoumine |
| Fondation pour la Recherche Médicale | DEQ20160334916 | Olivier Thoumine |
| Centre National de la Recherche Scientifique | | Olivier Thoumine |
| Commission Franco-Américaine Fulbright | | Olivier Thoumine |
| Conseil Régional Aquitaine | « SiMoDyn » | Olivier Thoumine |
| Investissements d'avenir | Labex BRAIN ANR-10-LABX-43 | Olivier Thoumine |
| France BioImaging | ANR-10INBS-04–01 | Olivier Thoumine |

The funders had no role in study design, data collection and interpretation, or the decision to submit the work for publication.

### Author contributions

Mathieu Letellier, Conceptualization, Data curation, Formal analysis, Funding acquisition, Investigation, Methodology; Matthieu Lagardère, Software, Validation; Béatrice Tessier, Data curation, Validation, Investigation, Methodology; Harald Janovjak, Conceptualization, Methodology; Olivier Thoumine, Conceptualization, Resources, Funding acquisition, Methodology, Project administration

### Author ORCIDs

Mathieu Letellier (iD) https://orcid.org/0000-0003-4008-298X
Olivier Thoumine (iD) https://orcid.org/0000-0002-8041-1349

### Ethics

Animal experimentation: The authors declare that they have complied with all relevant ethical regulations (study protocol approved by the Ethical Committee of Bordeaux CE50). Animals were raised in our animal facility; they were handled and killed according to European ethical rules.

Decision letter and Author response
Decision letter https://doi.org/10.7554/eLife.52027.sa1
Author response https://doi.org/10.7554/eLife.52027.sa2

## Additional files

### Supplementary files

- Source code 1. LTP program description.

- Transparent reporting form

### Data availability

All data generated or analysed during this study are included in the manuscript and supporting files.

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
