## [Decision Letter]

**Acceptance summary:**

The authors demonstrate the impact of an elegant technique, using light-induced phosphorylation of neuroligin-1 to gauge the impact of endogenous neuroligin-1 on synaptic function and plasticity in organotypic cultures of hippocampal CA1 neurons. This is an important advance for the field, answering fundamental issues and opening the door to new experiments to be pursued in the future.

**Decision letter after peer review:**

Thank you for submitting your article "Optogenetic control of excitatory post-synaptic differentiation through neuroligin-1 tyrosine phosphorylation" for consideration by *eLife*. Your article has been reviewed by three peer reviewers, and the evaluation has been overseen by a Reviewing Editor and Gary Westbrook as the Senior Editor. The following individual involved in review of your submission has agreed to reveal their identity: Markus Missler (Reviewer #1).

Summary:

Three reviewers have read, reviewed and discussed your work. All three appreciate the novelty and importance of the work, a view that is shared by the reviewing editor. The authors demonstrate the impact of light-induced phosphorylation of neuroligin-1 on synaptic function in organotypic cultures of hippocampal CA1 neurons. They report that Nlgn1 phosphorylation at residue Y782 enhances AMPA receptor mediated excitatory PSCs and increases dendritic spine density. In addition, they find reduced LTP levels upon phosphorylation, which they explain by trapping AMPAR in the synapse at the cost of depleting extrasynaptic AMPAR populations. Together, this study confirms key findings and concepts from previous papers (Giannone et al., 2013; Letellier et al., 2018) that examined manipulations of the Nlgn gene and protein. The strong point of the current manuscript is the successful use of an elegant, state-of-the-art method of light-induced phosphorylation and the fact that they address for the first time, at least in the majority of experiments, effects are observed in the background of endogenous Nlgn expression levels. All reviewers consider the work sufficiently novel and relevant for eventual publication in *eLife* if the authors are able to address major concerns outlined below. It is the general opinion of the reviewers that this additional work is within the scope of the two month window required by *eLife*, and would not necessitate generation of any new reagents. The additional work would provide essential controls for the final LTP experiments, and are therefore considered essential.

Essential Revisions:

1) The LTP data under optogenetically induced phosphorylation needs to contain at least the Nlgn1 KO or shRNA control under light condition to prove that abolishment of LTP is in fact due to the phosphorylation of Nlgn1. The authors have included this control for other phenotypic observations (AMPAR EPSCs, dendritic spines) but the LTP experiment should include at least one of these control conditions as well.

2) Figure 4B) It is not well explained why a statistical test for paired data sets can be used here since it does not represent the same cells before and after light induction.

3) In a previous paper, authors showed that overexpression of Nlg1 WT and Nlg1 Y782A but not Nlg1 Y782F affect AMPAR-EPSCs (Lettellier et al., 2018), but here in a replacement system, Nlg1 WT and Y782F do not have a significant effect on AMPAR EPSC. Can the authors discuss? Further, can the authors make comparisons with the previously published work more transparent?

4) Figure 3E and F. comparisons between different conditions should be added, or the authors can state why such comparisons are not necessary.

5) Nlg1 KO has been shown to decrease NMDAR-EPSC, but here the authors did not see a decrease. This discrepancy with the literature should be acknowledged and commented upon in the text.

---

## [Author Response]

Essential Revisions:1) The LTP data under optogenetically induced phosphorylation needs to contain at least the Nlgn1 KO or shRNA control under light condition to prove that abolishment of LTP is in fact due to the phosphorylation of Nlgn1. The authors have included this control for other phenotypic observations (AMPAR EPSCs, dendritic spines) but the LTP experiment should include at least one of these control conditions as well.

To answer this point, we performed LTP experiments in organotypic slices from Nlgn1 KO mice, under the same protocol as for slices from WT animals. We first verified using the pharmacological inhibitor AP5 that the elicited LTP was NMDAR-dependent (Figure 5—figure supplement 1A). Importantly, light stimulation of optoFGFR1 did not alter LTP in Nlgn1 KO neurons (Figure 5 C, D), indicating that Nlgn1 phosphorylation is responsible for the decreased LTP in wild type neurons. This is in agreement with the fact that light activation of optoFGFR1 does not affect basal EPSC amplitude in Nlgn1 KO neurons (Figure 3D, F), and is therefore not predicted to alter LTP according to our simulations.

However, we were surprised to find that Nlgn1 KO neurons exhibited similar LTP magnitude compared to control neurons from wild type slices, despite a decreased NMDA/AMPA ratio (**see response to major comment #5**). The discrepancy between these results and previous studies using Nlgn1 knock-out or knock-down approaches (Jiang et al., 2017; Wu et al., 2019; Budreck et al., 2013; Kim et al., 2008), is unclear and will require further investigation, which we think are beyond the scope of this study. Nevertheless, one or several possible explanations can be proposed:

1) the model (acute slices vs. organotypic slices): Studies using acute slices (e.g., Jiang et al., 2017; Kim et al., 2008; Wu et al., 2019) all found a full blockade of NMDAR-dependent LTP at CA3-CA1 synapses in the absence of Nlgn1. However, Budreck et al., 2013, reported in organotypic slices that NMDAR-LTP was not fully blocked, at least in the early stages (Figure 5: first 10-20 min after the stimulation).

2) the stimulation protocol (tetanus vs. pairing): We used a protocol similar to Budreck et al. (2 Hz stimulation with the post-synaptic cell @ 0 mV under voltage-clamp) but delivered 200 pulses instead of 100. Given that Budreck et al. found some potentiation in the initial stages of LTP (unlike studies using high frequency stimulations in acute slices), It is possible that our prolonged stimulation allowed to reach the threshold which is necessary to activate downstream signaling.

3) the neuronal maturation: Shipman et al., 2012, have shown that LTP induced by a tetanus at CA3-CA1 synapses is impaired in acute slices from young (P11-P14) but not mature animals (P40) when Nlgn1 is knocked-down with a miRNA-based approach. Although we performed our experiments in organotypic slices at DIV10-14, which would be closer to P11-P14, we cannot exclude that neuronal maturation is not altered in our slice model and affect the ability of neurons to undergo LTP.

2) Figure 5B) It is not well explained why a statistical test for paired data sets can be used here since it does not represent the same cells before and after light induction.

We decided to use the Wilcoxon signed-rank test to compare the LTP magnitude between electroporated neurons vs. control neighbors because EPSCs in these neurons were recorded simultaneously under the same conditions, i.e., the same input fibers were stimulated. We therefore think that these data can be considered as paired (see also previous studies using the Wilcoxon signed-rank test or the paired t-test to compare data from “paired whole cell recordings”: e.g., Shipman and Nicoll, Neuron 2012; Shipman et al., 2011; Watson et al., *eLife* 2017).

3) In a previous paper, authors showed that overexpression of Nlg1 WT and Nlg1 Y782A but not Nlg1 Y782F affect AMPAR-EPSCs (Letellier et al., 2018), but here in a replacement system, Nlg1 WT and Y782F do not have a significant effect on AMPAR EPSC. Can the authors discuss? Further, can the authors make comparisons with the previously published work more transparent?

The reviewer is right to ask for a more transparent comparison of this study which uses optogenetic activation of Nlgn1 phosphorylation, with our previous study (Letellier et al., 2018) which relied mostly on the use of tyrosine point mutants (Y782A/F). Indeed, upon over-expression in slices from Nlgn1 KO mice, Nlgn1-WT and Nlgn1-Y782A increase AMPAR-mediated EPSCs by about 4 fold, which is related to the increase in the number of synaptic contacts containing PSD-95 and AMPARs, while Nlgn1 Y782F does not cause such an increase (only 2-fold, significantly lower than Nlgn1-WT and -Y782A), most likely because it is unable to recruit a proper excitatory scaffold and AMPARs in front of newly recruited glutamatergic terminals.

Under replacement conditions (shRNA to Nlgn1 + rescue Nlgn1 in slices from WT mice), Nlgn1-WT did not increase AMPAR-mediated EPSCs, as expected from a full functional replacement. In contrast, Nlgn1-Y782A increased AMPAR-EPSCs by 4-fold and fully blocked LTP, as compared with non-electroporated neurons (Letellier et al., 2018), most likely resulting from a saturating recruitment of post-synaptic AMPARs at preexisting synapses and not from an increase in the number of synaptic contacts. In this respect, Y782A can be considered as a *gain-of function* mutation. Finally, the *loss-of-function* mutation Nlgn1-Y782F did not decrease AMPAR-mediated EPSCs as compared to non-electroporated neurons or neurons re-expressing Nlgn1-WT, as might have been expected (Letellier et al., 2018 - Figure 6F), potentially because of compensatory mechanisms independent of Nlgn1 in already assembled synapses. However, the non-phosphorylatable Nlgn1-Y782F mutant also partially blocked LTP (Letellier et al., 2018 - Figure 6G, H), suggesting that an optimal level of Nlgn1 phosphorylation is important for synapse potentiation. Indeed, we show here that photo-stimulation of optoFGFR1 increases basal AMPAR-EPSCs and partially occludes LTP. The fact that the optoFGFR1 is not as efficient as Nlgn1-Y782A in these processes can be due to the fact that: i) the Nlgn1 phosphorylation is not maximal, or ii) that the phosphorylation is not as potent in inhibiting gephyrin binding as the Y782A mutation. We modified the Discussion (third paragraph) to make this comparison more explicit.

4) Figure 4E and F. comparisons between different conditions should be added, or the authors can state why such comparisons are not necessary.

In addition to the Wilcoxon signed-rank test which shows a significant increase of EPSC amplitude in electroporated neurons expressing Nlgn1-WT and exposed to light relatively to their non electroporated counterparts, we have now added the comparisons between sets of paired data using the Kruskall-Wallis test, as requested by the reviewers (now Figure 4E). However, in contrast, to the data set comparing wild-type and knock-out slices in Figures 2D and 3F, we did not find any significant statistical difference between the conditions (WT vs. Y782F, light vs. dark). This might be explained by the higher variability in the data collected, which likely results from the replacement strategy in which the ratio between shRNA and recombinant Nlgn1 varies from cell to cell.

5) Nlg1 KO has been shown to decrease NMDAR-EPSC, but here the authors did not see a decrease. This discrepancy with the literature should be acknowledged and commented upon in the text.

We thank the reviewers for this comment which allows us to clarify our results. In agreement with previous reports (Chubykin et al., 2007; Kwon et al., 2012, Jiang et al., 2017), we do find a decrease in the NMDA-to-AMPA ratio when comparing Nlgn1 KO (or knocked-down) with WT (or rescued) neurons from different slices (see Figure 5—figure supplement 1B). However, this comparison is not apparent in the graph plots from Figures 2 and 3, where EPSC amplitudes from electroporated cells have been normalized to control cells from the same slice (i.e., from the same genotype) to isolate the effect of optoFGFR1 activation alone. Therefore, in Figure 2E, one should read that light activation of optoFGFR1 fails to affect NMDAR currents in the electroporated neurons relatively to its control (whether WT or KO).